# Oxidized glutathione reverts carbapenem resistance in $bla_{NDM-1}$-carrying *Escherichia coli*

Dongyang Ye [1,2], Xiaowei Li[1], Liang Zhao[1], Saiwa Liu[1], Xixi Jia[1], Zhinan Wang[1], Jingjing Du[1], Lirui Ge[1], Jianzhong Shen [1✉] & Xi Xia [1✉]

## Abstract

The emergence of drug-resistant Enterobacteriaceae carrying plasmid-mediated β-lactamase genes has become a significant threat to public health. Organisms in the Enterobacteriaceae family containing New Delhi metallo-β-lactamase-1 (NDM-1) and its variants, which are capable of hydrolyzing nearly all β-lactam antibacterial agents, including carbapenems, are referred to as superbugs and distributed worldwide. Despite efforts over the past decade, the discovery of an NDM-1 inhibitor that can reach the clinic remains a challenge. Here, we identified oxidized glutathione (GSSG) as a metabolic biomarker for $bla_{NDM-1}$ using a non-targeted metabolomics approach and demonstrated that GSSG supplementation could restore carbapenem susceptibility in *Escherichia coli* carrying $bla_{NDM-1}$ in vitro and in vivo. We showed that exogenous GSSG promotes the bactericidal effects of carbapenems by interfering with intracellular redox homeostasis and inhibiting the expression of NDM-1 in drug-resistant *E. coli*. This study establishes a metabolomics-based strategy to potentiate metabolism-dependent antibiotic efficacy for the treatment of antibiotic-resistant bacteria.

**Keywords** Carbapenems; Metabolomics; NDM-1; Oxidized Glutathione
**Subject Categories** Microbiology, Virology & Host Pathogen Interaction; Pharmacology & Drug Discovery

## Introduction

New Delhi metallo-β-lactamase-1 (NDM-1) can catalyze the hydrolysis of almost all β-lactams, including carbapenems (Yong et al, 2009). Gram-negative bacteria harboring plasmid-encoded NDM-1 have experienced rapid worldwide dissemination and have become a global threat (Hasan et al, 2012; Gao et al, 2020). Although continuous research has been conducted on the design and identification of NDM-1 inhibitors, none of the candidates that can overcome NDM-1 resistance have been approved for clinical use (Linciano et al, 2019). Recent evidence has revealed that

β-lactams are metabolism-dependent bactericidal antibiotics and that reactive metabolic byproducts induced by antibiotics play a crucial role in their bactericidal activity (Belenky et al, 2015). We hypothesized that NDM-1-positive bacteria undergo metabolic compensation under antibiotic pressure, and perturbing NDM-1-induced metabolic homeostasis may reverse their drug resistance.

## Results and discussion

To characterize the metabolic alterations induced by $bla_{NDM-1}$, we profiled the global metabolome of the resistant strain (pHSG398/$bla_{NDM-1}$) and susceptible strain (pHSG398) of *Escherichia coli* cultured in medium without antibiotics. Fifty-three differential putative metabolites were screened, of which 24 were upregulated, and 29 were downregulated, suggesting that the metabolic state of *E. coli* carrying $bla_{NDM-1}$ was significantly different from that of the susceptible strain (Fig. EV1). Notably, we observed a significant downregulation (21.9-fold) in oxidized glutathione (GSSG) in NDM-1-positive *E. coli*. In general, the ratio of reduced glutathione (GSH) to GSSG is considered an indicator of redox status (Masip et al, 2006). Downregulation of GSSG only in NDM-1-positive *E. coli* indicates a state of reductive stress in the resistant strain. To further explore the metabolic perturbations of $bla_{NDM-1}$-carrying *E. coli* under the pressure of carbapenem antibiotics, three carbapenems were used to stimulate the resistant strain *E. coli* (pHSG398/$bla_{NDM-1}$). Complex metabolic changes were observed after the stimulation of three carbapenem antibiotics, and 25 common differential metabolites represented major responses of $bla_{NDM-1}$-carrying *E. coli* to carbapenem treatment (Fig. 1A; Appendix Fig. S1; Appendix Tables S1–3). Among them, the concentrations of both GSH and GSSG decreased, but the suppressed levels of GSSG were much higher than that of GSH, leading to increased ratios of GSH/GSSG. After stimulation with meropenem, imipenem, and ertapenem, the GSH/GSSG ratio of $bla_{NDM-1}$-carrying *E. coli* increased by 2.92, 5.66, and 2.41 times, respectively. The metabolic pathway enrichment analysis revealed significant perturbations in various crucial metabolic pathways, including glutathione metabolism, pantothenate and CoA biosynthesis, purine metabolism, and nicotinate and nicotinamide metabolism, in $bla_{NDM-1}$-carrying *E. coli* (Appendix Fig. S2). Taken together, these metabolite alterations suggest that $bla_{NDM-1}$ confers drug

[1]National Key Laboratory of Veterinary Public Health and Safety, College of Veterinary Medicine, China Agricultural University, Beijing, China. [2]College of Veterinary Medicine, Northwest A&F University, Yangling, Shaanxi, China. ✉E-mail: sjz@cau.edu.cn; xxia@cau.edu.cn

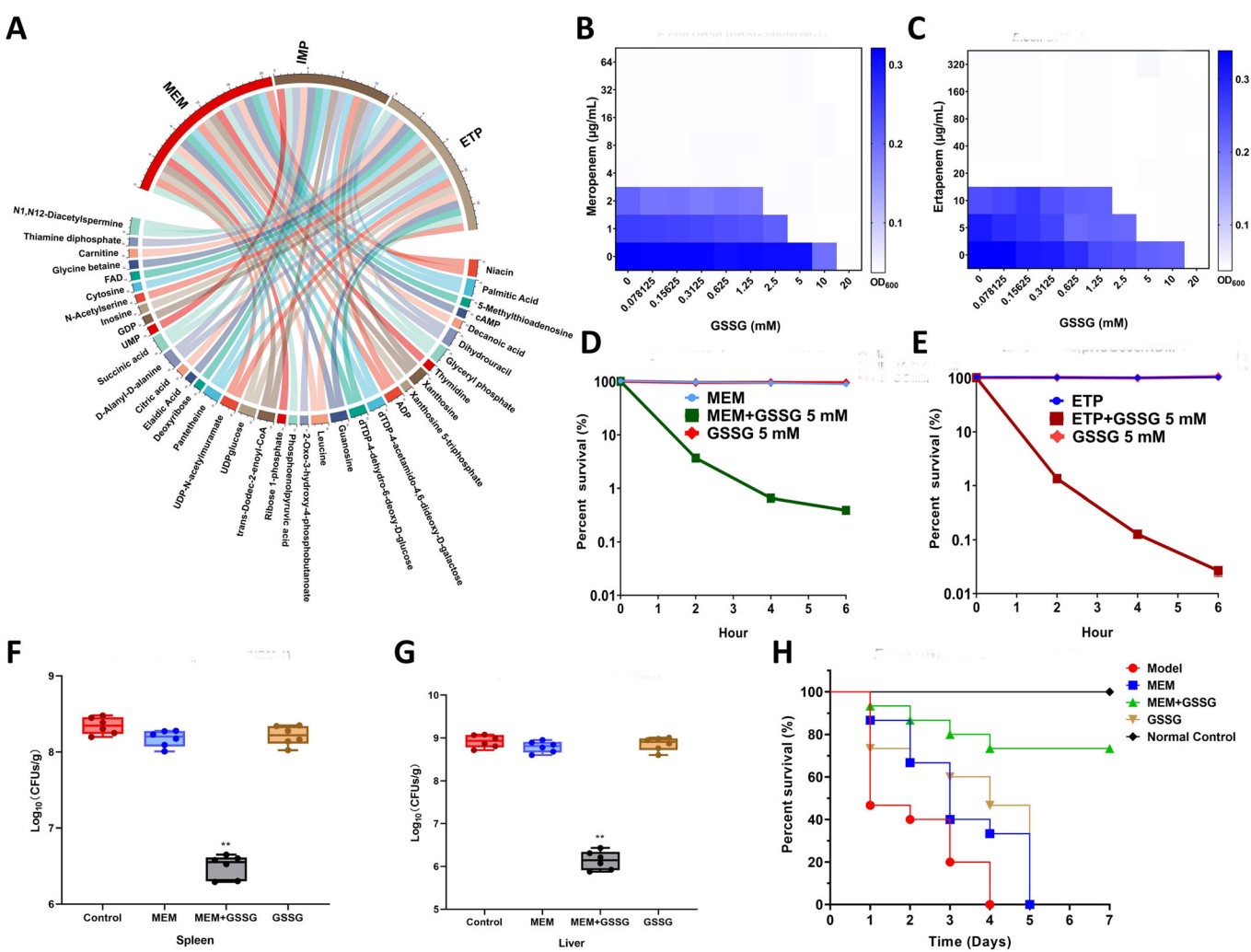

**Figure 1. In vitro and in vivo synergistic bactericidal effect of oxidized glutathione (GSSG) in combination with carbapenems.**

(A) Common differential metabolites of $bla_{NDM-1}$-carrying E. coli exposed to three carbapenem antibiotics ($n = 6$ biological replicates). (B, C) Checkerboard broth microdilution assays of GSSG combined with carbapenems against $bla_{NDM-1}$-carrying E. coli. (D, E) Survival of $bla_{NDM-1}$-carrying E. coli after treatment of GSSG, carbapenems, and combination therapy ($n = 3$ biological replicates). (F, G) Bacterial loads in the tissues of infected mice ($n = 6$ per group). Box plots represent the median with interquartile range, and the whiskers indicate the minimum and maximum values. One-way ANOVA analysis, $**p < 0.01$. (H) Survival rates of infected mice after treatment of GSSG, meropenem, and combination therapy ($n = 15$ per group). Source data are available online for this figure.

resistance through a dual mechanism. In addition to the hydrolysis of β-lactam antibiotics, $bla_{NDM-1}$ induces an antioxidant metabolic state to attenuate the lethality of antibiotics.

To test whether GSSG can be used as an adjuvant for carbapenem antibiotics, the synergistic bactericidal effect of GSSG combined with carbapenems was evaluated on E. coli DH5α (pHSG398/$bla_{NDM-1}$) and a wild-type strain E. coli 20SC1DM33 carrying $bla_{NDM-1}$. For E. coli DH5α (pHSG398/$bla_{NDM-1}$), when the concentration of GSSG was >2.5 mM, the minimum inhibitory concentration (MIC) of meropenem and ertapenem reduced by four times, and the fractional inhibitory concentration (FICI) was <0.5, indicating good synergistic antibacterial effect of GSSG with meropenem and ertapenem (Fig. 1B,C). Similarly, for E. coli 20SC1DM33, when the concentration of GSSG was >5 mM, the decreased fold change in MIC of meropenem or ertapenem combined with GSSG could reach eight times, and the FICI was

also <0.5. Furthermore, we assessed the synergistic effect of GSSG with meropenem on various $bla_{NDM-1}$ variants, including $bla_{NDM-4}$, $bla_{NDM-5}$, $bla_{NDM-7}$, and $bla_{NDM-9}$. The combination of GSSG and meropenem reduced the MIC of E. coli carrying $bla_{NDM-4}$ and $bla_{NDM-5}$ by 16 times and that of E. coli carrying $bla_{NDM-7}$ and $bla_{NDM-9}$ by eight times, suggesting a common mechanism of GSSG-mediated drug susceptibility recovery (Fig. EV2). In addition, we performed time-kill assays to assess the synergistic effects of GSSG and carbapenems. The survival rates of E. coli DH5α (pHSG398/$bla_{NDM-1}$) and E. coli 20SC1DM33 were nearly 100% when treated alone with 5 mM GSSG, 1 μg/mL meropenem, and 2 μg/mL ertapenem. However, for E. coli DH5α (pHSG398/$bla_{NDM-1}$), when meropenem or ertapenem combined with GSSG for 2 h, the survival rate was <5% (3.68 ± 0.136% and 1.35 ± 0.144%, respectively), and then <1% (0.39 ± 0.016% and 0.03 ± 0.005%, respectively) for 6 h (Fig. 1D,E). For E. coli 20SC1DM33, the

survival rate decreased to $7.05 \pm 0.51\%$ and $1.30 \pm 0.029\%$ when exposed to meropenem + GSSG and ertapenem + GSSG for 2 h, respectively. The survival rate of the drug-resistant strain was <0.2% after 6 h of combined treatment with GSSG and carbapenems (Fig. EV3).

To further investigate the potential efficacy of GSSG in vivo, we assessed the feasibility of combination therapy in resentitizing resistant strains to antibiotics in a mouse model of intraperitoneal infection with a lethal dose of *E. coli* DH5α (pHSG398/$bla_{NDM-1}$) or *E. coli* 20SC1DM33. GSSG or meropenem monotherapy had little effect on the bacterial load of $bla_{NDM-1}$-carrying *E. coli* in the tissues after 24 h. In contrast, when infected with *E. coli* DH5α (pHSG398/$bla_{NDM-1}$), the combinatorial treatment group exhibited a > tenfold reduction in spleen bacterial load and approximately three orders of magnitude reduction in liver bacterial load compared to the control group (Fig. 1F,G). Similar results were observed in mice infected with *E. coli* 20SC1DM33, in which bacterial loads in both the liver and spleen were reduced by three orders of magnitude after treatment with meropenem + GSSG (Fig. EV4). In a separate experiment, all mice infected with *E. coli* DH5α (pHSG398/$bla_{NDM-1}$) in the non-treated group died within 4 days, and meropenem and GSSG monotherapy failed to rescue any infected mice within 5 days. However, the survival rate of the combination therapy increased to 73% 5 days after infection (Fig. 1H). A similar survival rate (67%) after co-therapy with meropenem and GSSG was obtained in mice infected with *E. coli* 20SC1DM33 (Fig. EV5).

The excessive accumulation of reactive oxygen species (ROS) in cells has been implicated as an important cause of bactericidal antibiotics (Kohanski et al, 2007; Foti et al, 2012). We investigated whether the addition of GSSG would affect the production of cellular ROS to promote the killing capabilities of carbapenem antibiotics against drug-resistant *E. coli*. The results showed that meropenem alone could not induce increased accumulation of ROS in drug-resistant strains, suggesting that antioxidant responses occurred in *E. coli* carrying $bla_{NDM-1}$ (Fig. 2A,B). GSSG increased the level of intracellular ROS, but to a lesser extent than hydrogen peroxide. Treatment with meropenem + GSSG led to significantly increased ROS levels compared to the non-treatment group ($p < 0.01$), showing an increase of more than twofold in *E. coli* DH5α (pHSG398/$bla_{NDM-1}$) and nearly 1.5-fold in *E. coli* 20SC1DM33. Next, we used three antioxidants (ascorbic acid, thiourea, and N-acetylcysteine) to test whether ROS inhibition abolished the synergistic effect of GSSG and carbapenems. In the presence of ascorbic acid, thiourea, and N-acetylcysteine, the survival rate of drug-resistant strains increased markedly after 6 h of culture treated with the combination of GSSG and meropenem ($p < 0.01$) (Fig. 2C,D).

Given that ROS inhibition restored the partial survival of drug-resistant bacteria, we hypothesized that GSSG supplementation would affect the hydrolysis of carbapenem antibiotics by NDM-1. To test this hypothesis, we first examined the effect of GSSG on the expression of $bla_{NDM-1}$. Elevated expression of $bla_{NDM-1}$ was observed in drug-resistant strains under the pressure of meropenem to resist the bactericidal effects of antibiotics (Fig. 2E,F). GSSG alone decreased the expression of $bla_{NDM-1}$ in *E. coli* DH5α (pHSG398/$bla_{NDM-1}$), but the reduction was significant when GSSG was combined with meropenem ($p < 0.05$) and was more pronounced in the wild-type resistant strain *E. coli* 20SC1DM33

($p < 0.01$). Western blotting was performed to confirm the expression of β-lactamase, which was consistent with the qPCR results (Fig. 2G,H). Furthermore, we quantified the levels of meropenem in drug-resistant strains with or without GSSG supplementation using liquid chromatography-tandem quadrupole mass spectrometry. As predicted, the combination of GSSG and meropenem led to a significant increase ($p < 0.01$) in the intracellular concentration of meropenem in the drug-resistant strains, which was 29.7-fold in *E. coli* DH5α (pHSG398/$bla_{NDM-1}$) and 10.6-fold in *E. coli* 20SC1DM33 (Fig. 2I,J).

Given the understanding of the structure and hydrolysis mechanism of NDM-1, valuable progress in addressing the challenge of drug-resistant bacteria carrying $bla_{NDM-1}$ mainly lies in the discovery of molecules that can interact with the active site of the enzyme (Feng et al, 2017). Several recent studies have suggested that the cellular metabolic state of bacteria can influence the efficacy of antibiotics. β-lactams are strongly metabolism-dependent, and ROS contribute, in part, to their lethality (Lobritz et al, 2015; Stokes et al, 2019). In this study, we characterized the metabolic profile of *E. coli* carrying $bla_{NDM-1}$ and identified suppressed levels of GSSG as a biomarker of metabolic modulation. GSSG supplementation re-sensitized NDM-1-positive *E. coli* to carbapenems by promoting ROS production, which potentiates the bactericidal ability of antibiotics. In addition, GSSG can inhibit metallo-β-lactamase expression, which in turn increases the intracellular level of antibiotics. The potential of combinatorial therapy with GSSG and carbapenems has been demonstrated in a mouse model of intraperitoneal infection. Our work proposes an alternative strategy to screen for metabolic adjuvants that can restore existing antibiotic lethality to address the clinical challenge of NDM-1.

## Methods

### Bacterial strains and cultivation

The bacterial strains used in this study included *E. coli* DH5α (pHSG398), *E. coli* DH5α (pHSG398/$bla_{NDM-1}$), and wild-type *E. coli* 20SC1DM33. All bacterial strains were cultured in Luria–Bertani (LB) medium at 37 °C.

### Metabolic profiling

Non-targeted metabolomics analyses were performed as previously described (Ye et al, 2021). Briefly, bacteria were quenched with methanol/ethylene glycol (45:55, v/v, −60 °C) and washed twice with 0.85% NaCl. The quenched pellets were extracted with boiling ethanol/water (75:25, v/v, 95 °C). The intracellular metabolites were separated using a reversed-phase column and a hydrophilic interaction column and detected using a time-of-flight mass spectrometer (Ab Sciex, 6600 +, Framingham, MA, USA) in positive and negative electrospray ionization modes, respectively. The raw data were preprocessed using Progenesis QI v.2.3 (Nonlinear Dynamics, Newcastle, UK) for baseline correction, alignment, and peak picking. The detected features were filtered (fold change >1.5, $p < 0.05$, CV < 30%, VIP > 1) to identify differential metabolites. Metabolites were putatively identified using database search against the ECMDB (http://www.ecmdb.ca).

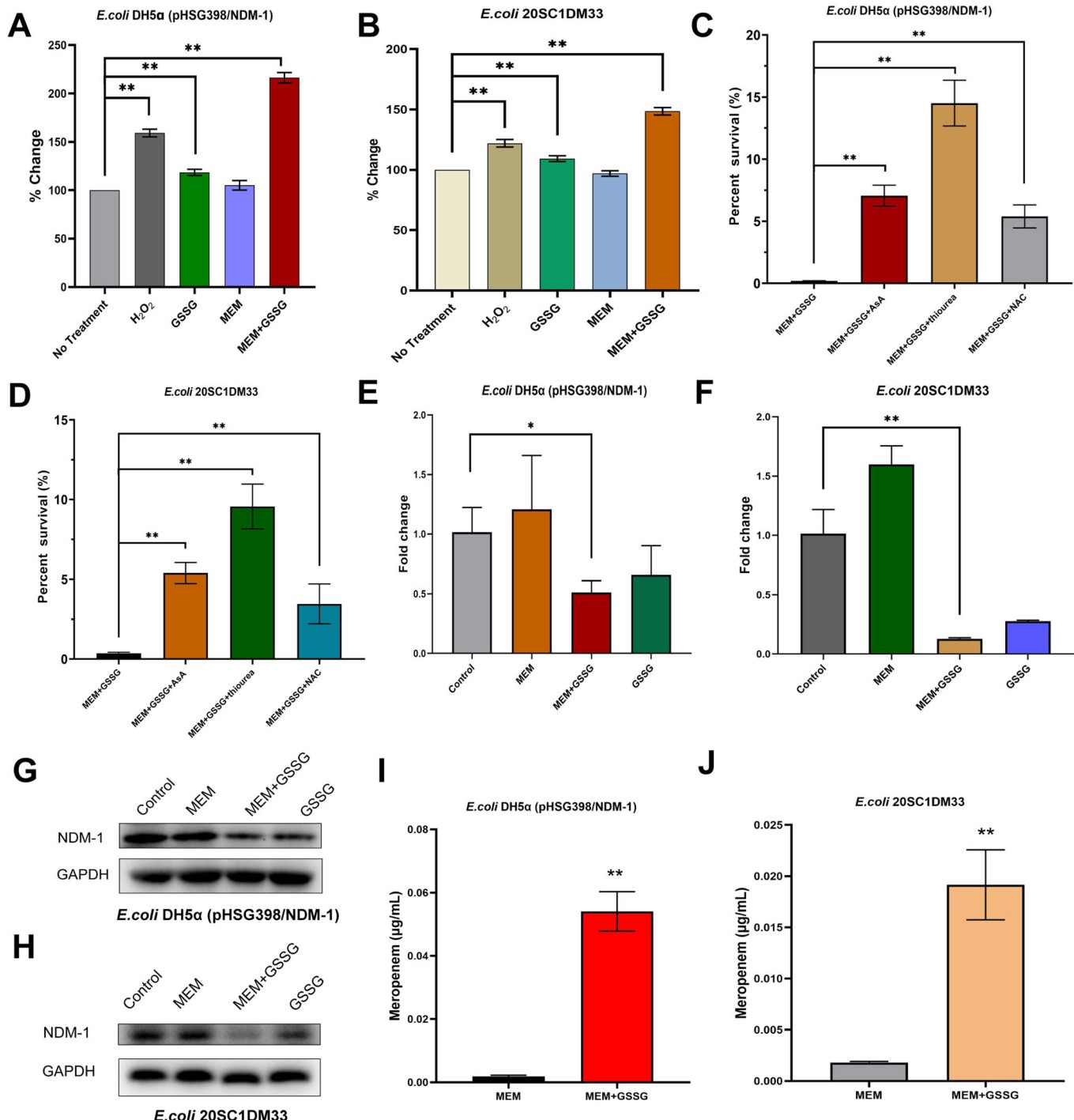

**Figure 2. Mechanism of oxidized glutathione (GSSG) in combination with carbapenem antibiotics.**

(A, B) GSSG + meropenem induces reactive oxygen species (ROS) in $bla_{NDM-1}$-carrying *E. coli* ($n = 3$ biological replicates). (C, D) Survival of $bla_{NDM-1}$-carrying *E. coli* treated by the combination of GSSG + meropenem and ROS inhibition ($n = 3$ biological replicates). (E, F) Transcription analysis of $bla_{NDM-1}$ in resistant strains treated with GSSG, meropenem, and combination therapy ($n = 3$ biological replicates). (G, H) Expression of NDM-1 decreased after the treatment of GSSG + meropenem. (I, J) Intracellular concentrations of meropenem in $bla_{NDM-1}$-carrying *E. coli* ($n = 3$ biological replicates). Data Information: Data are presented as mean ± SD. One-way ANOVA analysis, *$p < 0.05$, **$p < 0.01$. Source data are available online for this figure.

Multivariate analysis was performed using MetaboAnalyst 5.0 (https://www.metaboanalyst.ca) and EZinfo software (Waters, Milford, MA, USA).

## Synergistic bacteriostasis experiment

MICs of GSSG and antibiotics (meropenem and ertapenem) for the tested strains were determined using the checkerboard microdilution method. Briefly, GSSG or antibiotics were serially diluted twofold in a sterile 96-well microliter plate with Mueller Hinton broth, and then the bacterial suspension ($1 \times 10^6$ CFU/mL, 100 μL) was added. The MICs were determined as the lowest concentrations of GSSG or antibiotics with no visible bacterial growth after 16–24 h of co-incubation at 37 °C. The checkerboard microdilution method was performed using a twofold serial dilution of compounds ($8 \times 10$ matrix) mixed with bacterial suspension ($5 \times 10^5$ CFU/mL) to assess the synergies between GSSG and antibiotics against drug-resistant strains. The fractional inhibitory concentration index (FICI) was determined using the following formula: FICI = (MIC $_{\text{compounds in combination}}$/MIC $_{\text{compound alone}}$) + (MIC $_{\text{antibiotics in combination}}$/MIC $_{\text{antibiotics alone}}$). A synergistic effect was defined as a FICI ≤ 0.5 (Gomara and Ramon-Garcia, 2019).

## Time-killing curve

GSSG (5 mM), antibiotics, or GSSG + antibiotics were cultured with bacterial suspensions ($1 \times 10^6$ CFU/mL) in M9 minimal medium supplemented with 10 mM ammonium acetate, 1 mM MgSO$_4$, and 100 μM CaCl$_2$. The concentrations of meropenem were 1 μg/mL for *E. coli* DH5α (pHSG398/NDM-1) and 2 μg/mL for *E. coli* 20SC1DM33. The concentrations of ertapenem were 5 μg/mL for *E. coli* DH5α (pHSG398/NDM-1) and 10 μg/mL for *E. coli* 20SC1DM33. At the time points 0, 2, 4, and 6 h, culture supernatants from different treatments were serially diluted, and then spotted onto LB agar. The bacterial colony counts were determined, and the percent survival was calculated.

## Determination of intracellular ROS

Single colonies of *E. coli* DH5α (pHSG398/NDM-1) and *E. coli* 20SC1DM33 were picked, inoculated into sterile LB broth, and incubated overnight at 37 °C. Overnight cultures were transferred to conical flasks containing LB broth and grown to an optical density (OD$_{600}$) of 0.35. The samples were centrifuged at 8000 rpm for 5 min, washed twice with 30 mL sterile saline, and resuspended in M9 minimal medium supplemented with 10 mM ammonium acetate, 1 mM MgSO$_4$, and 100 μM CaCl$_2$. Subsequently, meropenem was added to the medium at a concentration of 1 μg/mL for *E. coli* DH5α (pHSG398/NDM-1) and 2 μg/mL for *E. coli* 20SC1DM33, followed by GSSG at a concentration of 50 mM. H$_2$O$_2$ (20 mM) was used as a positive control. Three replicates were prepared for each group and incubated at 37 °C for 2 h. The level of intracellular ROS was determined using a ROS high-quality fluorescence assay kit (Genmed Scientifics Inc., Shanghai, China).

## ROS inhibition

*E. coli* DH5α (pHSG398/NDM-1) and *E. coli* 20SC1DM33 were cultured at 37 °C until the exponential phase was reached. The

cultures were adjusted to a McFarland turbidity of 0.5, and then diluted 1:100 in 10 mL M9 media to ~$10^6$ CFU/mL. Meropenem was added to the media at concentrations of 1 μg/mL for *E. coli* DH5α (pHSG398/NDM-1) and 2 μg/mL for *E. coli* 20SC1DM33, followed by GSSG at a concentration of 50 mM. ROS inhibitors, including ascorbic acid, thiourea, and N-acetylcysteine were added to the medium at a concentration of 50 mM. The bacterial survival rate was determined after exposure to meropenem, GSSG, and the inhibitor for 6 h.

## qRT-PCR

Total RNA was extracted using a HiPure Bacterial RNA Kit (Guangzhou Meiji Biotechnology Co., Ltd., Guangzhou, China) and quantified by the ratio of absorbance (260 nm/280 nm) using a Nanodrop spectrophotometer (Thermo Fisher Scientific, Waltham, MA, USA). qRT-PCR was performed using PowerUp SYBR Green Master Mix (Thermo Fisher). Cycling conditions consisted of an initial denaturation step at 95 °C for 3 min, followed by 40 cycles at 95 °C for 30 s, 60 °C for 30 s, and 72 °C for 30 s, and then 72 °C for 10 min. All samples were analyzed in triplicate, and the 16 S rRNA gene was used as an endogenous control, as described in our previous publication. Fold changes in gene expression were analyzed using the $2^{-\Delta\Delta CT}$ method. The primer sequences used in the qRT-PCR analysis are listed in Appendix Table S4.

## Western blot analysis

Bacterial cultures were collected and boiled at 100 °C for 10 min after ultrasonication. Each sample was fractionated by sodium dodecyl sulfate-polyacrylamide gel electrophoresis and transferred onto a polyvinylidene difluoride membrane. Next, the membrane was incubated with primary polyclonal anti-NDM-1 antibody (1:1000; NBP1-77688; NOVUS Biologicals, Littleton, CO, USA) and subsequently incubated with secondary antibody (1:1000; anti-rabbit IgG, HRP-linked antibody; #7074; Cell Signaling Technology, Inc., Danvers, MA, USA). Finally, the membranes were visualized with an enhanced chemiluminescence substrate using a Tanon-4200 gel imaging analysis system (Tianneng Technology Co., Ltd, Shanghai, China).

## Detection of intracellular antibiotic

The fortified concentrations of meropenem were 10 μg/mL for *E. coli* DH5α (pHSG398/NDM-1) and 48 μg/mL for *E. coli* 20SC1DM33, and the concentration of GSSG was 50 mM. Three biological replicates were prepared for each group and the cultures were centrifuged after 2 h of treatment. The pelleted cells were washed twice with 0.85% NaCl, resuspended in 1 mL 30% methanol/water, and lysed on ice with ultrasonication. The samples were centrifuged, and the supernatants were injected into a UHPLC-tandem quadrupole mass spectrometer (Qtrap 6500 + , Ab Sciex). The LC separation was performed on a BEH Shield RP C$_{18}$ (Waters) maintained at 40 °C. The injection volume was 5 μL and the flow rate was 0.4 mL/min. The mobile phases consisted of solvent A (0.1% formic acid in water) and solvent B (0.1% formic acid in methanol). The gradient elution program was as follows: 0–0.5 min, 99% A; 0.5–1.5 min, 99–70% A; 1.5–2.0 min, 70% A; 2.0–3.0 min, 70-10% A; 3.0–3.5 min, 10% A; 3.5-3.6 min, 10-99% A;

**The paper explained**

**Problem**

The excessive use of antibiotics has resulted in the emergence and wide dissemination of multidrug-resistant bacteria carrying antibiotic-resistance genes, particularly $bla_{NDM-1}$, which encodes metallo-beta-lactamase NDM-1. This enzyme can confer resistance to nearly all beta-lactam antibiotics, including carbapenems, posing a serious threat to public health and safety. However, the discovery of an effective NDM-1 inhibitor remains underexplored.

**Results**

A non-targeted metabolomics approach was employed to identify altered metabolic profiles in *Escherichia coli* carrying $bla_{NDM-1}$. When exposed to carbapenems, the intracellular metabolic state shifted toward antioxidant stress, suggesting that $bla_{NDM-1}$ not only leads to antibiotic hydrolysis but also inhibits bactericidal effects due to antibiotic-induced oxidative stress. Metabolic reprogramming through exogenous oxidized glutathione (GSSG) was found to reduce intracellular ROS production and inhibit the expression of $bla_{NDM-1}$. The combined bactericidal effect of GSSG and carbapenems against $bla_{NDM-1}$-positive *E. coli* strains was demonstrated through infection experiments in mice.

**Impact**

Metabolomics analysis results indicate that $bla_{NDM-1}$ mediates carbapenem resistance via a dual resistance mechanism. The combination of GSSG and carbapenems provides an alternative treatment for $bla_{NDM-1}$-carrying *E. coli* infections. Metabolic reprogramming in combination with antibiotics is a promising strategy for combating multidrug-resistant bacteria.

and 3.6–5.0 min, 99% A. The target compound was detected by multiple reaction monitoring in the positive electrospray ionization mode. The transitions of meropenem were as follows: $m/z$ 384.2 > 141.1 (transition for quantification), $m/z$ 384.2 > 254.2 (transition for quantification).

## Mouse infection assays

To determine the bacterial load in tissues, 24 male Balb/c mice (6 weeks old) were intraperitoneally infected with 200 μL of bacterial suspension ($1 \times 10^9$ CFU/mL), which resulted in 100% mortality at 168 h post infection, and then randomly divided into four groups. One hour after infection, each group of mice was treated with saline, meropenem (5 mg/kg), GSSG (150 mg/kg), or meropenem (5 mg/kg) + GSSG (150 mg/kg). All mice were euthanized 24 h post infection, and liver and spleen tissues were collected to measure bacterial load (CFU/g) by plate counts. In a separate experiment, 60 male Balb/c mice (6 weeks old) were intraperitoneally infected with 200 μL of bacterial suspension ($1 \times 10^9$ CFU/mL) and randomly divided into four groups. Fifteen mice were injected with saline as the negative control group. Infected mice were treated with saline, meropenem (5 mg/kg), GSSG (150 mg/kg), and meropenem (5 mg/kg) + GSSG (150 mg/kg). The survival rates of all groups were monitored every 24 h. The mice were provided by Beijing Vital River Laboratory Animal Technology Co. (Beijing, China). The laboratory animal usage license was certified by the Beijing Association for Science and Technology (SYXK-20210012). Animal protocols were approved by the Committee on Animal Welfare and Ethics of China Agricultural University.

## Statistical analysis

The investigator was not blinded to the experimental conditions, and none of the mice were excluded from the experiments. Statistical significance was determined using an unpaired two-sided Student's $t$ tests or one-way ANOVA. The levels of significance were as follows: $*P < 0.05$, $**P < 0.01$, and $***P < 0.001$.

## For more information

Please see https://ngdc.cncb.ac.cn/.

## Data availability

Raw metabolomic data have been deposited at the China National Genomics Data Center of Open Archive for Miscellaneous Data (https://ngdc.cncb.ac.cn/omix/releaseList) under accession number OMIX005757.

## Peer review information

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

## Acknowledgements

This work was supported by the National Natural Science Foundation of China (32072916, 32302928).

## Author contributions

**Dongyang Ye**: Formal analysis; Investigation; Methodology; Writing—original draft. **Xiaowei Li**: Formal analysis; Investigation; Methodology. **Liang Zhao**: Investigation; Visualization. **Saiwa Liu**: Formal analysis; Investigation. **Xixi Jia**: Formal analysis; Investigation. **Zhinan Wang**: Formal analysis; Validation. **Jingjing Du**: Investigation; Methodology. **Lirui Ge**: Investigation; Methodology. **Jianzhong Shen**: Supervision; Project administration. **Xi Xia**: Conceptualization; Supervision; Funding acquisition; Project administration; Writing—review and editing.

## Disclosure and competing interests statement

The authors declare no competing interests.

# Expanded View Figures

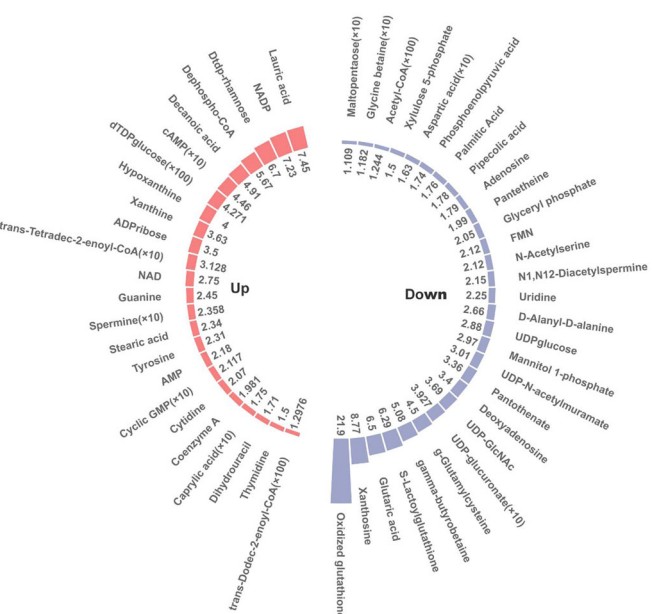

**Figure EV1.** *bla*~NDM-1~ **induces broad metabolic alterations in *E. coli*.**

Metabolomics analysis of *E. coli* carrying pHSG398 and pHSG398/*bla*~NDM-1~ was performed using liquid chromatography-high resolution mass spectrometry. Upregulated and downregulated metabolites in drug-resistant strain are indicated with red and blue, respectively.

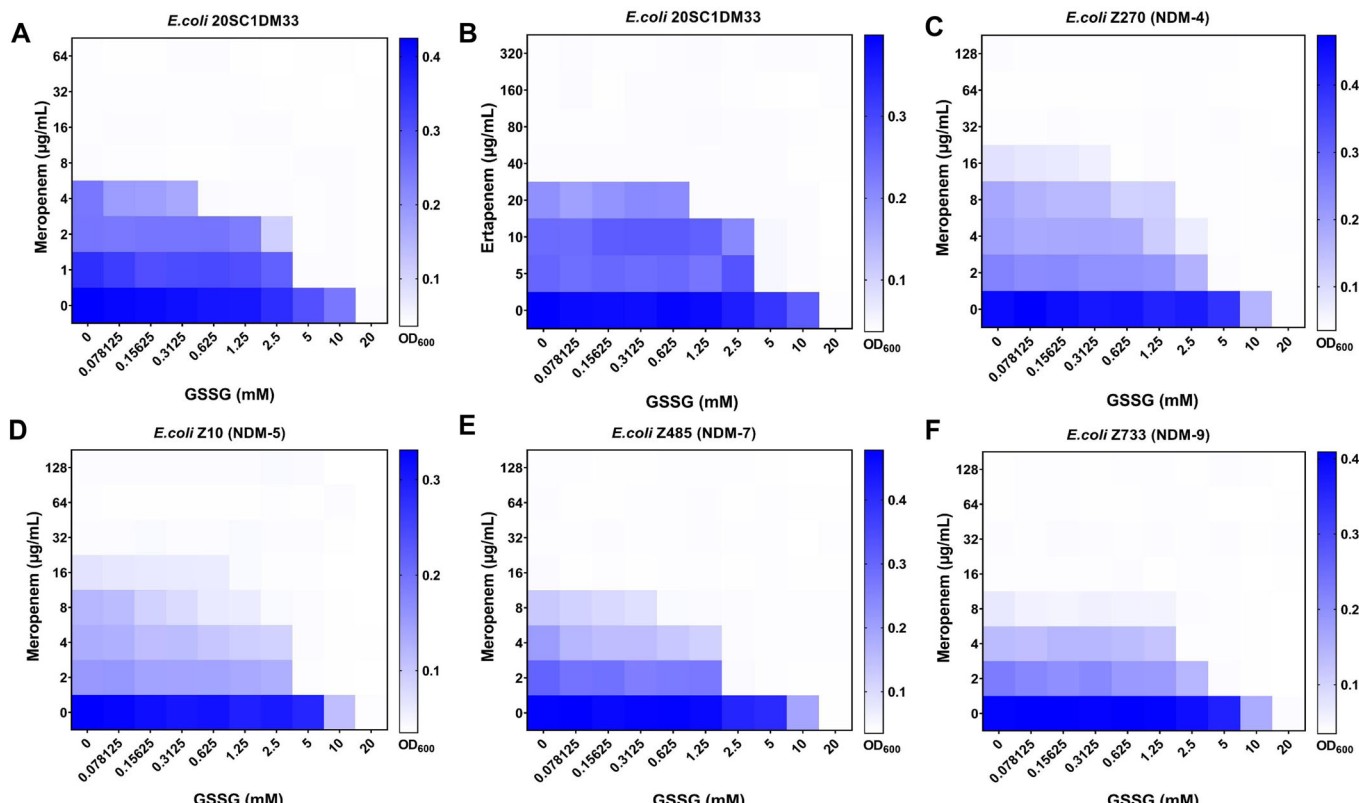

**Figure EV2. Synergistic effect of GSSG combined with carbapenems on wild-type resistant strains *E. coli* carrying *bla*<sub>NDM-1</sub> and variants.**

(A) *E. coli* 20SC1DM33 treated with meropenem and GSSG. (B) *E. coli* 20SC1DM33 treated with ertapenem and GSSG. (C) *E. coli* Z270 carrying *bla*<sub>NDM-4</sub> treated with meropenem and GSSG. (D) *E. coli* Z10 carrying *bla*<sub>NDM-5</sub> treated with meropenem and GSSG. (E) *E. coli* Z485 carrying *bla*<sub>NDM-7</sub> treated with meropenem and GSSG. (F) *E. coli* Z733 carrying *bla*<sub>NDM-9</sub> treated with meropenem and GSSG.

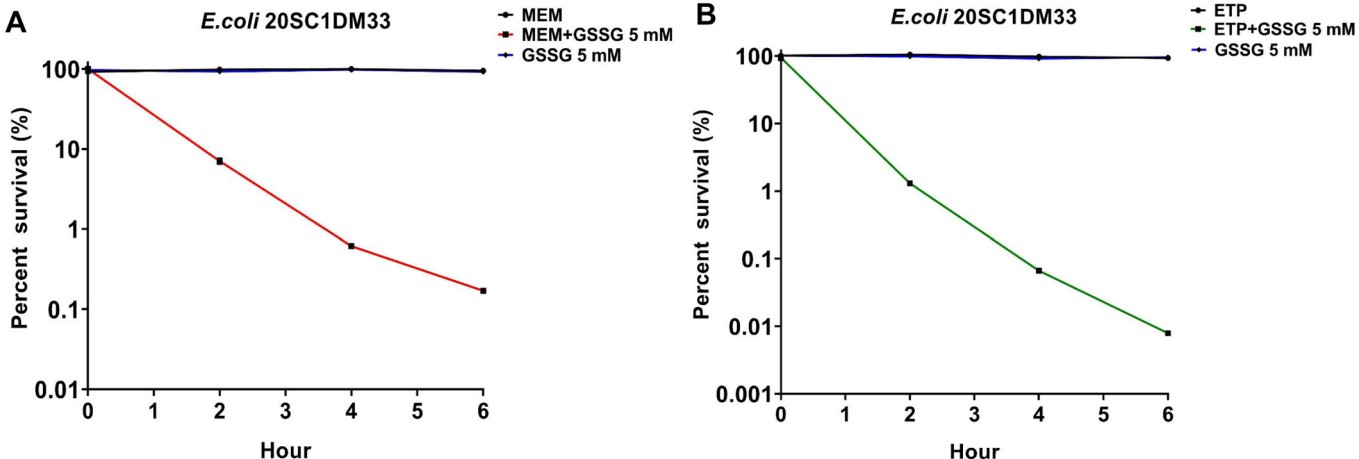

**Figure EV3. In vitro synergistic bactericidal effect of oxidized glutathione (GSSG) in combination with carbapenems on a wild-type strain *E. coli*.**

(A) Survival of *E. coli* 20SC1DM33 after treatment of GSSG, meropenem (MEM), and combination therapy ($n = 3$ biological replicates). (B) Survival of *E. coli* 20SC1DM33 after treatment of GSSG, ertapenem (ETP), and combination therapy ($n = 3$ biological replicates).

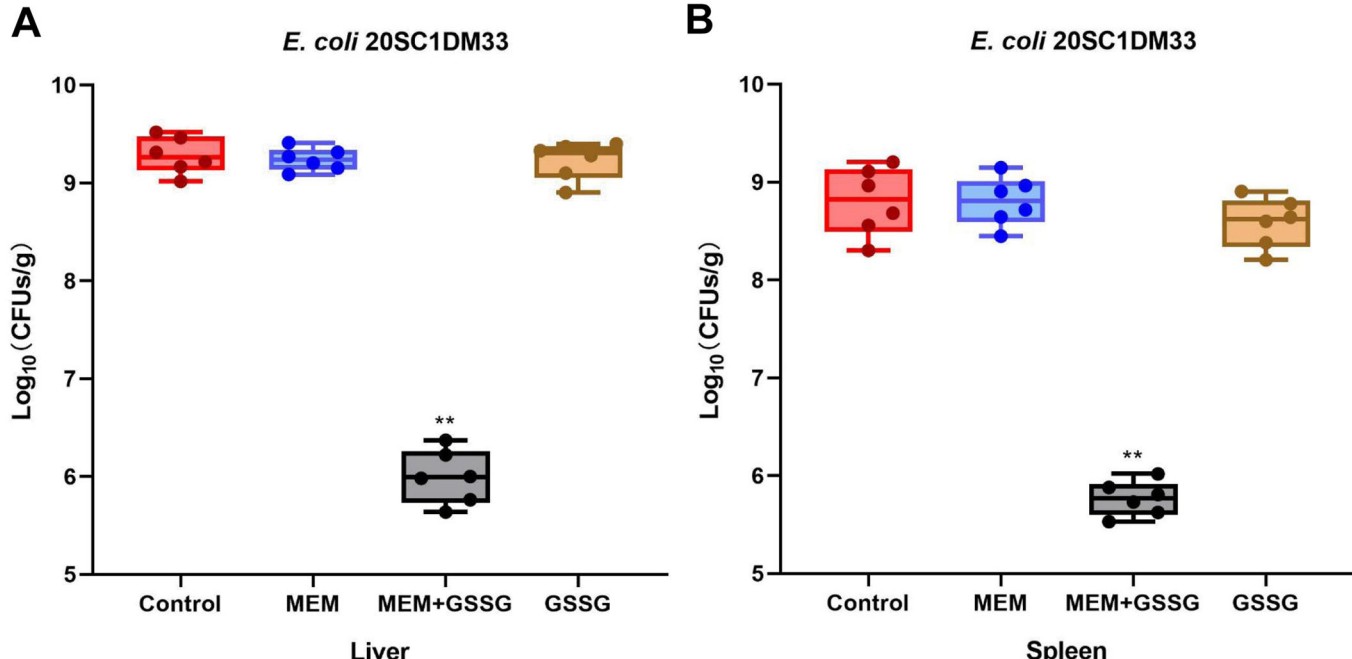

**Figure EV4. Bacterial loads in the tissues of mice infected with a wild-type resistant strain *E. coli*.**

(A) Bacterial loads in the liver of infected mice ($n = 6$ per group) after treatment of oxidized glutathione (GSSG), meropenem (MEM), and combination therapy.
(B) Bacterial loads in the spleen of infected mice ($n = 6$ per group) after treatment of GSSG, MEM, and combination therapy. Data Information: Box plots represent the median with interquartile range, and the whiskers indicate the minimum and maximum values. One-way ANOVA analysis, $**p < 0.01$.

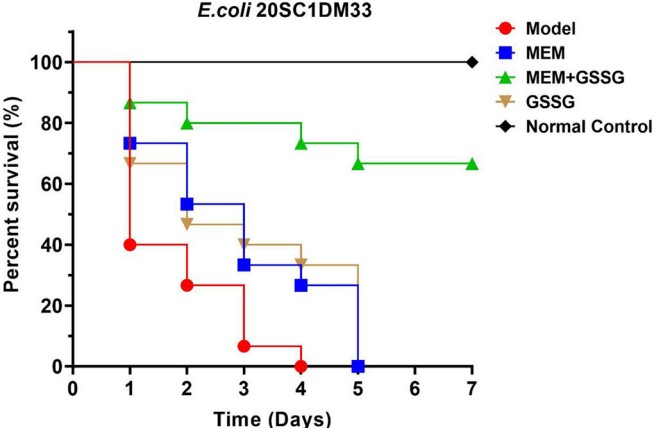

**Figure EV5.** **In vivo synergistic bactericidal effect of oxidized glutathione (GSSG) in combination with meropenem (MEM) on a wild-type strain _E. coli._**

Survival plot of infected mice ($n = 15$ per group) after treatment of GSSG, MEM, and combination therapy.

