## [Peer Review File · EMBO Molecular Medicine]

Oxidized glutathione reverts carbapenem resistance in blaNDM-1-carrying Escherichia coli

Dong-Yang Ye, Xiaowei Li, Liang Zhao, Saiwa Liu, Xixi Jia, Zhinan Wang, Jingjing Du, Lirui Ge, Jianzhong Shen, and Xi Xia

Corresponding authors: Xi Xia (xxia@cau.edu.cn) , Jianzhong Shen (sjz@cau.edu.cn)

Review Timeline:

Submission Date:	21st Dec 23
Editorial Decision:	11th Jan 24
Revision Received:	2nd Feb 24
Editorial Decision:	5th Mar 24
Revision Received:	13th Mar 24
Accepted:	15th Mar 24

Editor: Zeljko Durdevic

Transaction Report:

11th Jan 2024

Dear Dr. Xia,

Thank you for the submission of your manuscript to EMBO Molecular Medicine. We have now received feedback from the two reviewers who agreed to evaluate your manuscript. Both referees recognize potential interest of the study but also raise important criticism that should be addressed in a major revision. If you would like to discuss further the points raised by the referees, I am available to do so via email or video. Let me know if you are interested in this option.

We would welcome the submission of a revised version within three months for further consideration. Please let us know if you require longer to complete the revision.

I look forward to receiving your revised manuscript.

Yours sincerely,

Zeljko Durdevic

We require:

- 1) A .docx formatted version of the manuscript text (including legends for main figures, EV figures and tables). Please make sure that the changes are highlighted to be clearly visible.
- 2) Individual production quality figure files as .eps, .tif, .jpg (one file per figure). For guidance, download the 'Figure Guide PDF': (<https://www.embopress.org/page/journal/17574684/authorguide#figureformat>).
- 3) A .docx formatted letter INCLUDING the reviewers' reports and your detailed point-by-point responses to their comments. As part of the EMBO Press transparent editorial process, the point-by-point response is part of the Review Process File (RPF), which will be published alongside your paper.
- 4) A complete author checklist, which you can download from our author guidelines (<https://www.embopress.org/page/journal/17574684/authorguide#submissionofrevisions>). Please insert information in the checklist that is also reflected in the manuscript. The completed author checklist will also be part of the RPF.
- 5) Please note that all corresponding authors are required to supply an ORCID ID for their name upon submission of a revised manuscript.
- 6) It is mandatory to include a 'Data Availability' section after the Materials and Methods. Before submitting your revision, primary datasets produced in this study need to be deposited in an appropriate public database, and the accession numbers and

database listed under 'Data Availability'. Please remember to provide a reviewer password if the datasets are not yet public (see <https://www.embopress.org/page/journal/17574684/authorguide#dataavailability>).

13) Author contributions: You will be asked to provide CRediT (Contributor Role Taxonomy) terms in the submission system. These replace a narrative author contribution section in the manuscript.

14) A Conflict of Interest statement should be provided in the main text.

15) Every published paper now includes a 'Synopsis' to further enhance discoverability. Synopses are displayed on the journal

webpage and are freely accessible to all readers. They include a short stand first (maximum of 300 characters, including space) as well as 2-5 one-sentences bullet points that summarizes the paper. Please write the bullet points to summarize the key NEW findings. They should be designed to be complementary to the abstract - i.e. not repeat the same text. We encourage inclusion of key acronyms and quantitative information (maximum of 30 words / bullet point). Please use the passive voice. Please attach these in a separate file or send them by email, we will incorporate them accordingly.

Please also suggest a striking image or visual abstract to illustrate your article as a PNG file 550 px wide x 300-800 px high.

**** Reviewer's comments ****

Referee #1 (Remarks for Author):

Emergence of novel antimicrobial resistance imposes a great concern globally. However, development of antimicrobials has come into a standstill phase. To cope with the challenge of AMR, novel strategies of combating against superbugs are urgently required. In this paper, the authors used an untargeted metabolomics approach to screen for microbial metabolites that reverse carbapenem resistance in *E. coli* carrying blaNDM-1. The use of metabolic reprogramming methods is currently a cutting-edge research area in curbing bacterial drug resistance. This research demonstrate a good research paradigm and exemplify Oxidized glutathione as an candidate to reverse the carbapenem resistance in *E.coli*. There are still some issues that need to be addressed before the manuscript can be accepted for publication.

1. What are the advantages of screening for drug resistance reversal metabolites by metabolic changes over high-throughput screening from large compound libraries?
2. Why did the authors choose only GSSG and not other differential metabolites to explore their ability to reverse bacterial resistance?
3. Are upregulated compounds valuable for screening for metabolites that reverse drug resistance?
4. Why used both C18 and HILIC modes for data acquisition in metabolomics analysis? What are the advantages over lower sample throughputs?
5. The results of the metabolic pathway enrichment analyses should be presented, both for the NDM's and for the drug-stimulated group, to better demonstrate pathway changes other than the glutathione metabolic pathway.

Referee #2 (Comments on Novelty/Model System for Author):

To determine the bacterial load in tissues, Balb/c mice were intraperitoneally infected with bacterial suspension. And the 251 mice were provided by Beijing Vital River Laboratory Animal Technology Co. (Beijing, 252 China). Laboratory animal usage license was certified by the Beijing Association for 253 Science and Technology (SYXK-20210012). Animal protocols were approved by the 254 Committee on Animal Welfare and Ethics of China Agricultural University

Referee #2 (Remarks for Author):

The manuscript reports that GSSG restores susceptibility to carbapenems in *E. coli* strains harboring the resistant gene blaNDM-1, suggesting the potential for combination therapy. The authors demonstrated the synergistic bactericidal effect of GSSG with carbapenems in vitro and in vivo, and investigated the related molecular mechanisms. NDM is one of the most important drug resistance genes, and all potential therapeutic approaches to it are of great importance. I recommend moderate revisions before acceptance for publication.

1. The authors should explain the principles of selecting strains for comparative metabolomics research, and why not wild resistant strains?
2. It is not particularly clear on what basis the GSSG was chosen, simply because of the fold change?
3. All differential metabolites under the pressure of three carbapenems should be presented, at least in the SI.
4. Have you investigated the effects of non-common differential metabolites in addition to the 25 common differential metabolites?
5. You mentioned that blaNDM-1 may mediate the antioxidant stress of host bacteria under the stimulation of carbapenems. What are the specific changes of the GSH/GSSG ratio?

Reviewer #1:**1. What are the advantages of screening for drug resistance reversal metabolites by metabolic changes over high-throughput screening from large compound libraries?**

High-throughput screening of active compounds from large compound libraries is a conventional approach for the discovery of antibiotic adjuvants. This approach is often laborious and time-consuming, yielding very few promising compounds, or even thousands of screens with no results at all. In our study, the screening of potential compounds focused on bacterial endogenous metabolites. Differential metabolites associated with drug-resistant genes reflect the metabolic compensation that must be paid for the expression of drug-resistant genes. Discovering antibiotic adjuvants from differential metabolites has a smaller screening range and a higher likelihood of finding effective compounds than screening with large compound libraries.

2. Why did the authors choose only GSSG and not other differential metabolites to explore their ability to reverse bacterial resistance?

It has been demonstrated that bactericidal antibiotics induce complex redox alterations that contribute to cellular damage and death, thus supporting an expanded model of antibiotic lethality. The metabolic state of bacteria also influences their susceptibility to antibiotics. The ratio of GSH to GSSG is an indicator of redox status. Therefore, the dramatic decrease in GSSG levels in drug-resistant strains immediately attracted us to conduct in-depth research. We will verify the synergistic effect of other differential metabolites in reversing drug resistance in future work.

3. Are upregulated compounds valuable for screening for metabolites that reverse drug resistance?

Both up- and down-regulated differential metabolites caused by drug-resistant gene should be considered as potential compounds. However, in this study, we focused on discovering antibiotic adjuvants from downregulated differential metabolites, as we could investigate their synergistic effect in metabolic reprogramming through exogenous supplementation of metabolites.

4. Why used both C18 and HILIC modes for data acquisition in metabolomics

analysis? What are the advantages over lower sample throughputs?

C18 columns generally provide good chromatographic separation for non-polar and moderately polar compounds, but their chromatographic retention is poor for polar compounds. On the contrary, HILIC columns are suitable for the analysis of polar compounds. Endogenous metabolites of bacteria are numerous and chemically diverse. The combined use of C18 and HILIC columns provided a relatively comprehensive bacterial metabolic profile. We prefer not to miss important potential antibiotic adjuvants compared to lower sample throughputs.

5. The results of the metabolic pathway enrichment analyses should be presented, both for the NDM's and for the drug-stimulated group, to better demonstrate pathway changes other than the glutathione metabolic pathway.

We have included enrichment analysis figures in the Appendix Figure S2 to better demonstrate pathway changes beyond the glutathione metabolic pathway (L 54-57).

Reviewer #2:

1. The authors should explain the principles of selecting strains for comparative metabolomics research, and why not wild resistant strains?

Comparative metabolomics studies should obtain biological samples that contain only univariable. In this study, we constructed two engineered strains, *E.coli* DH5 α (pHSG398/*bla*_{NDM-1}) and *E.coli* DH5 α (pHSG398), to ensure that the *bla*_{NDM-1} gene served as the single variable. The genetic background of wild strains remains unclear, and there may be coexistence of other drug-resistant genes in addition to the *bla*_{NDM-1} gene, which would affect the relevance of the bacterial metabolic profile to the target drug-resistant gene.

2. It is not particularly clear on what basis the GSSG was chosen, simply because of the fold change?

First, the level of GSSG decreased dramatically in *bla*_{NDM-1}-positive *E. coli*. Second, GSSG is an important redox marker, and it has been shown that the redox status within bacteria has a significant impact on the bactericidal effect of antibiotics. Finally, the results of metabolic pathway analysis indicated that glutathione metabolism was the main metabolic pathway perturbed in sensitive versus resistant bacteria as well as

in drug-resistant bacteria under the pressure of carbapenems.

3. All differential metabolites under the pressure of three carbapenems should be presented, at least in the SI.

All differential metabolites under the pressure of three carbapenems were presented in the Appendix Table S1-S3.

4. Have you investigated the effects of non-common differential metabolites in addition to the 25 common differential metabolites?

In this study, we conducted stimulation experiments on *bla*_{NDM-1}-positive bacteria using meropenem, imipenem, and ertapenem, resulting in the identification of 25 common differential metabolites. By analyzing these common differential metabolites, we investigated the major metabolic pathways disrupted by carbapenems in *bla*_{NDM-1}-positive bacteria to determine potential metabolic reprogramming strategies. Furthermore, each of the three carbapenems also led to corresponding specific differential metabolites. However, we did not evaluate these non-common differential metabolites, and further work is warranted to investigate their potential effects in restoring susceptibility of *bla*_{NDM-1}-positive *E. coli* to carbapenems.

5. You mentioned that *bla*_{NDM-1} may mediate the antioxidant stress of host bacteria under the stimulation of carbapenems. What are the specific changes of the GSH/GSSG ratio?

We have added specific changes in the GSH/GSSG ratio under the stimulation of carbapenems in the manuscript (L 52-54).

5th Mar 2024

Dear Dr. Xia,

Thank you for the resubmission of your manuscript to EMBO Molecular. I am pleased to inform you that we will be able to accept your manuscript pending the following final amendments:

1) Figures: EV figure should also be uploaded as separate file, therefore, please submit individual, high-resolution file for each EV figure. Please provide more detailed description of the EV figures and Appendix figures in the legends (e.g. similar as you have for figure EV2 each figure should have a title, text that explains the figure, abbreviations explained etc.) For more information on figure presentation please check "Author Guidelines".

<https://www.embopress.org/page/journal/17574684/authorguide#datapresentationformat>

2) In the main manuscript file, please do the following:

- Please address all comments suggested by our data editors listed below:

o Figure legends:

1. Please note that a separate 'Data Information' section is required in the legends of figures 2a-f, i-j; EV 4a-b.

2. Please note that individual figure legends for figures EV 4a-b are not provided in the manuscript. This needs to be rectified.

3. Please indicate the statistical test used for data analysis in the legends of figures 1f-g; 2a-f, i-j; EV 4a-b.

4. Please note that the box plots need to be defined in terms of minima, maxima, centre, bounds of box and whiskers, and percentile in the legends of figures 1f-g; EV 4a-b.

5. Although 'n' is provided, please describe the nature of entity for 'n' in the legends of figures 2a-f, i-j.

6. Please note that the error bars are not defined in the legends of figures 2a-f, i-j.

- Add callouts for Appendix Tables 1-4.

- Rename "Methods" to "Materials and Methods".

- Author contributions: Please remove it from the manuscript and specify author contributions in our submission system. CRediT has replaced the traditional author contributions section because it offers a systematic machine-readable author contributions format that allows for more effective research assessment. You are encouraged to use the free text boxes beneath each contributing author's name to add specific details on the author's contribution. More information is available in our guide to authors:

<https://www.embopress.org/page/journal/17574684/authorguide#authorshipguidelines>

3) Appendix: Add page numbers in the table of content.

4) Synopsis:

- Synopsis image: Please select one of the two provided images for visual abstract, resize the image to 550 px-wide x (250-400)-px high and upload it as a high-resolution jpeg file.

5) For more information: This space should be used to list relevant web links for further consultation by our readers. Could you identify some relevant ones and provide such information as well? Some examples are patient associations, relevant databases, OMIM/proteins/genes links, author's websites, etc...

6) As part of the EMBO Publications transparent editorial process initiative (see our Editorial at

<http://embomolmed.embopress.org/content/2/9/329>), EMBO Molecular Medicine will publish online a Review Process File (RPF) to accompany accepted manuscripts. This file will be published in conjunction with your paper and will include the anonymous referee reports, your point-by-point response and all pertinent correspondence relating to the manuscript. Let us know whether you agree with the publication of the RPF and as here, if you want to remove or not any figures from it prior to publication. Please note that the Authors checklist will be published at the end of the RPF.

7) Please provide a point-by-point letter INCLUDING my comments as well as the reviewer's reports and your detailed responses (as Word file).

I look forward to reading a new revised version of your manuscript as soon as possible.

Yours sincerely,

Zeljko Durdevic

*** Instructions to submit your revised manuscript ***

- 1) a .docx formatted version of the manuscript text (including Figure legends and tables)
 - 2) Separate figure files*
 - 3) supplemental information as Expanded View and/or Appendix. Please carefully check the authors guidelines for formatting Expanded view and Appendix figures and tables at <https://www.embopress.org/page/journal/17574684/authorguide#expandedview>
 - 4) a letter INCLUDING the reviewer's reports and your detailed responses to their comments (as Word file).
 - 5) The paper explained: EMBO Molecular Medicine articles are accompanied by a summary of the articles to emphasize the major findings in the paper and their medical implications for the non-specialist reader. Please provide a draft summary of your article highlighting
 - the medical issue you are addressing,
 - the results obtained and
 - their clinical impact.This may be edited to ensure that readers understand the significance and context of the research. Please refer to any of our published articles for an example.
 - 6) For more information: There is space at the end of each article to list relevant web links for further consultation by our readers. Could you identify some relevant ones and provide such information as well? Some examples are patient associations, relevant databases, OMIM/proteins/genes links, author's websites, etc...
 - 7) Author contributions: the contribution of every author must be detailed in a separate section.
 - 8) EMBO Molecular Medicine now requires a complete author checklist (<https://www.embopress.org/page/journal/17574684/authorguide>) to be submitted with all revised manuscripts. Please use the checklist as guideline for the sort of information we need WITHIN the manuscript. The checklist should only be filled with page numbers where the information can be found. This is particularly important for animal reporting, antibody dilutions (missing) and exact values and n that should be indicated instead of a range.
 - 9) Every published paper now includes a 'Synopsis' to further enhance discoverability. Synopses are displayed on the journal webpage and are freely accessible to all readers. They include a short stand first (maximum of 300 characters, including space) as well as 2-5 one sentence bullet points that summarise the paper. Please write the bullet points to summarise the key NEW findings. They should be designed to be complementary to the abstract - i.e. not repeat the same text. We encourage inclusion of key acronyms and quantitative information (maximum of 30 words / bullet point). Please use the passive voice. Please attach these in a separate file or send them by email, we will incorporate them accordingly.
- You are also welcome to suggest a striking image or visual abstract to illustrate your article. If you do please provide a jpeg file 550 px-wide x 300-800px high.
- 10) A Conflict of Interest statement should be provided in the main text
 - 11) Please note that we now mandate that all corresponding authors list an ORCID digital identifier. This takes <90 seconds to complete. We encourage all authors to supply an ORCID identifier, which will be linked to their name for unambiguous name

identification.

Currently, our records indicate that the ORCID for your account is 0000-0002-9630-7909.

Link Not Available

Photos 400-800 DPI

*Additional important information regarding figures and illustrations can be found at

<https://bit.ly/EMBOPressFigurePreparationGuideline>. See also figure legend preparation guidelines:

<https://www.embopress.org/page/journal/17574684/authorguide#figureformat>

***** Reviewer's comments *****

Referee #1 (Comments on Novelty/Model System for Author):

The revised manuscript has been improved, and no further comments here.

Referee #1 (Remarks for Author):

The current format is suitable for this story. A full research article is not necessary.

Referee #2 (Remarks for Author):

Is suitable for publication.

The authors addressed the minor editorial issues.

15th Mar 2024

Dear Dr. Xia,

We are pleased to inform you that your manuscript is accepted for publication and is now being sent to our publisher to be included in the next available issue of EMBO Molecular Medicine.
